# Reduced Carbon Dioxide by Overexpressing *EPSPS* Transgene in Arabidopsis and Rice: Implications in Carbon Neutrality through Genetically Engineered Plants

**DOI:** 10.3390/biology13010025

**Published:** 2023-12-31

**Authors:** Li-Xue Sun, Ning Li, Ye Yuan, Ying Wang, Bao-Rong Lu

**Affiliations:** 1Ministry of Education Key Laboratory for Biodiversity and Ecological Engineering, School of Life Sciences, Fudan University, Songhu Road 2005, Shanghai 200438, China; 17110700023@fudan.edu.cn (L.-X.S.); yuany19@fudan.edu.cn (Y.Y.); 2State Key Laboratory of Genetic Engineering, School of Life Sciences, Fudan University, Songhu Road 2005, Shanghai 200438, China; 17110700073@fudan.edu.cn

**Keywords:** Arabidopsis, biotechnology, carbon fixation, gene expression, genetic engineering, photosynthesis, rice

## Abstract

**Simple Summary:**

Overexpressing the *EPSPS* gene may increase the consumption of ambient CO_2_ by plants. To explore the relationship between CO_2_ consumption by plants and expression of the *EPSPS* gene, we measured the expression level of the *EPSPS* gene, intercellular CO_2_ concentration, expression of global genes, and photosynthetic ratios in genetically engineered (GE) and non-GE Arabidopsis and rice plants. Our results showed substantial increases in *EPSPS* gene expression, CO_2_ consumption, expression of genes in carbon-fixation related pathways, and photosynthetic ratios. The results indicate that overexpressing the *EPSPS* gene can enhance CO_2_ consumption by GE plants with increased expression of genes in carbon-fixation related pathways. The findings not only expand our knowledge on the relationship between overexpression of the *EPSPS* gene(s) and carbon fixation, but also provide a novel strategy to reduce global CO_2_ for carbon neutrality, in addition to increases in crop production by genetic engineering of the *EPSPS* gene(s) in plants.

**Abstract:**

With the increasing challenges of climate change caused by global warming, the effective reduction of carbon dioxide (CO_2_) becomes an urgent environmental issue for the sustainable development of human society. Previous reports indicated increased biomass in genetically engineered (GE) Arabidopsis and rice overexpressing the 5-enolpyruvylshikimate-3-phosphate synthase (*EPSPS*) gene, suggesting the possibility of consuming more carbon by GE plants. However, whether overexpressing the *EPSPS* gene in GE plants consumes more CO_2_ remains a question. To address this question, we measured expression of the *EPSPS* gene, intercellular CO_2_ concentration, photosynthetic ratios, and gene expression (RNA-seq and RT-qPCR) in GE (overexpression) and non-GE (normal expression) Arabidopsis and rice plants. Results showed substantially increased *EPSPS* expression accompanied with CO_2_ consumption in the GE Arabidopsis and rice plants. Furthermore, overexpressing the *EPSPS* gene affected carbon-fixation related biological pathways. We also confirmed significant upregulation of four key carbon-fixation associated genes, in addition to increased photosynthetic ratios, in all GE plants. Our finding of significantly enhanced carbon fixation in GE plants overexpressing the *EPSPS* transgene provides a novel strategy to reduce global CO_2_ for carbon neutrality by genetic engineering of plant species, in addition to increased plant production by enhanced photosynthesis.

## 1. Introduction

The great challenge of global climate change caused by the gradual increases in greenhouse gases (GHG) has posed tremendous threats to the global economic security and sustainable development of human society [1]. Carbon dioxide (CO_2_), particularly that produced from various anthropogenic activities, constitutes an important GHG in the world [1]. In response to the challenge posed by excessive GHG emissions, the global community has reached the Paris Climate Agreement [2,3] to take decisive actions and make significant strides toward the development of a low-carbon economy. In addition, scientific investigations are carried out to explore effective resolutions for achieving global carbon reduction or carbon neutrality to maintain the continued world development. These resolutions include applying clean energy [4,5,6], conserving energy and reducing CO_2_ emission [7,8], regulating carbon sink in the ecosystems [9,10,11,12], and increasing the terrestrial vegetation coverage [13].

As an important carbon pool, the terrestrial ecosystems fix ~30% global atmospheric CO_2_ emitted by anthropogenic activities [14,15]. Consequently, terrestrial ecosystems have played an important role in reducing the atmospheric CO_2_, which is critical for maintaining the stability of the global climate [14]. In terrestrial ecosystems, forests cover ~30% of the total terrestrial area [15,16], contributing a larger fraction to the carbon sink, meaning CO_2_ fixation by organisms [15,17,18]. Grasslands, croplands (crop fields), and other terrestrial floral ecosystems are also important for global terrestrial CO_2_ sink [15,18]. Usually, different types of stored carbon (sink) in plants are in the form of biomass that is transformed from solar energy and CO_2_ through photosynthesis [15].

Photosynthesis is a pivotal source for carbon sink in terrestrial ecosystems [19] due to its capacity of converting atmospheric CO_2_ into the fixed carbon by plants [20]. Given that the further increase in terrestrial vegetation is relatively difficult, we can significantly increase the efficiency of photosynthesis and carbon fixation by green plants. It is reported that the efficiency of photosynthesis can determine the possibility for plants to absorb or consume CO_2_, thereby impacting global climate change [21]. As a result, terrestrial plants have great potential to extensively increase the efficiency of photosynthesis and carbon fixation at the global scale. Thus, enhancing photosynthetic efficiency of terrestrial plants provides an ideal strategy for carbon neutrality to achieve GHG CO_2_ reduction. In addition, enhancing photosynthesis can also increase crop production, thereby contributing to food security.

Previous studies indicated that genetically engineered (GE) plants overexpressing the exogenous or endogenous *EPSPS* genes substantially increased biomass in rice [22,23,24] and *Arabidopsis*
*thaliana* [25,26]. In addition, overexpressing the *EPSPS* gene also significantly increased photosynthetic ratios [22]. These results together suggest that GE plants overexpressing the *EPSPS* gene may consume more atmospheric CO_2_ to meet the demand for the increased photosynthesis and biomass. The *EPSPS* gene, encoding the essential enzyme in the shikimate pathway, is present in all plant species, as well as in many microorganisms [27]. Furthermore, the *EPSPS* gene is also important for the biosynthesis of metabolites such as aromatic amino acids, lignin, and auxin [28]. Therefore, it is possible to use green plants to increase the ratios of photosynthesis and CO_2_ fixation through genetical engineering.

Whether overexpression of the *EPSPS* gene would generally increase the consumption of atmospheric CO_2_ by GE plants is still unclear, but presents interesting questions in terms of its practical implications for carbon neutrality. To address this question, we investigated the intercellular CO_2_ concentration of Arabidopsis and rice plants overexpressing the *EPSPS* gene produced through genetic engineering [22,25], in addition to their photosynthetic ratios. We also applied the RNA-seq and RT-qPCR molecular techniques to determine the possible impact of the *EPSPS* transgene on the global genes, and particularly carbon-fixation related genes, in Arabidopsis or rice plants. The objectives of this study were to answer the following questions: (1) Does overexpression of the *EPSPS* gene increase the consumption or fixation ratios of CO_2_ in the GE experimental plants? (2) What are the changes of the gene expression pattern globally in the experimental plants caused only by overexpressing the *EPSPS* gene? (3) What is the impact of overexpressing the *EPSPS* gene on the carbon-fixation related genes, as well as photosynthetic ratios in the GE experimental plants? Answers to these questions will facilitate our understanding for designing novel strategies to reduce atmospheric CO_2_ by overexpressing the *EPSPS* genes in GE plants as the carbon sink, in addition to increases in crop production through enhanced plant photosynthesis.

## 2. Materials and Methods

### 2.1. Plant Materials and Growth Conditions

Two GE Arabidopsis (*Arabidopsis thaliana*) lineages (coded as A1 and A2) overexpressing the *EPSPS* transgene (cloned from cultivated rice) and their corresponding non-GE counterparts were used in this study to represent the dicotyledonous plants. The Arabidopsis plants were the 4th generation (T_4_) derived from the GE and non-GE homozygous T_3_ Arabidopsis, respectively [25]. The GE Arabidopsis lineages, containing a single copy of the *EPSPS* gene driven by the CaMV 35S promoter, were generated through Agrobacterium-mediated transformation. The GE Arabidopsis plants were proven to tolerate the glyphosate herbicide [25]. A total of 204 Arabidopsis plants were included in the experiment with the following greenhouse growth conditions: 22 °C all day, 16-h light, 8-h dark.

One GE rice (*Oryza sativa*) lineage (coded as W) overexpressing the same rice *EPSPS* transgene and its corresponding non-GE counterpart were used to represent the monocotyledonous plants. The rice plants were derived from the hybrids between an *EPSPS* GE rice line and a weedy rice population at the 4th generation (F_4_) from Wang’s experiment [22]. The GE rice lineage, containing a single copy of the *EPSPS* gene driven by the ubiquitin (Ubi) promoter (from maize), was generated through Agrobacterium-mediated transformation. The GE rice plants also tolerated the glyphosate herbicide [22]. A total of 42 rice plants were included in the experiment with the following greenhouse growth conditions: 30 °C during the day, 27 °C during the night, 8-h light, 16-h dark. Sampling and the measurement were made on 30-day-old Arabidopsis plants and 35-day-old rice plants with fully expanded leaves.

### 2.2. Gas-Exchange Measurements

Carbon fixation ratios were estimated by the intercellular CO_2_ concentration (Ci) and photosynthetic ratios using a portable photosynthetic system equipped with the multiphase flash fluorometer and chamber (LI-6800, LI-COR, Inc., Logan, NE, USA). To ensure the accurate measurement, all gas-exchange measurements were conducted on three plants for each Arabidopsis or rice lineage at a constant atmospheric CO_2_ concentration (Ca) with 400 μmol mol^−1^, gas flow rate of 500 μmol s^−1^, fan speed of 10,000 rpm, a leaf temperature of 25 °C, and 60% relative humidity (all controlled by the LI-6800). All the experimental leaves were sufficiently large enough to cover the entire window area of the equipment. The leaves were clamped on the window area for ~10 min before the measured parameters being stable, then the formal measurement started. To determine the light-saturated photosynthetic ratio of Arabidopsis plants, the photosynthetic measurement was made using a series of light intensities at the 1500, 1200, 800, and 500 μmol m^−2^ s^−1^. The light-saturated photosynthetic ratio and the intercellular CO_2_ concentration (Ci) was determined within ten minutes at photosynthetic photon flux density (PPFD) of 1200 μmol m^−2^ s^−1^ for Arabidopsis. In addition, a PPFD of 1000 μmol m^−2^ s^−1^ was used for measuring Ci and photosynthetic ratio of rice plants within ten minutes following the Wang’s description [22]. Further, we used the Ca-Ci value to indicate the amount of atmospheric CO_2_ consumption and fixation during the photosynthetic assimilation. The calculation of Ca-Ci value is based on the differences between measured air CO_2_ concentrations (Ca) and intercellular CO_2_ concentrations (Ci).

### 2.3. RNA-Sequencing and Analysis

Two GE Arabidopsis lineages overexpressing the *EPSPS* gene and two non-GE counterparts were selected for RNA-sequencing (RNA-seq). Each lineage included five biological replicates, each consisting of six Arabidopsis plants, and were randomly sampled from fully expanded leaves. Extraction of total RNA, reverse transcription, and adaptor ligation for hybridization were performed using the NEBNext UltraTM RNA Library Prep Kit for Illumina (NEB, Ipswich, MA, USA) for cDNA library construction. A total of 20 cDNA libraries were subjected to RNA-seq analysis using an Illumina novaseq6000 with a 150 bp read length and the sequencing was conducted using pair-end cDNA libraries. Our study had a sequence depth of ~20 million uniquely mapped reads per sample using the following procedure. First, adaptor sequences and low-quality sequences were trimmed using the Trimmomatic. Second, the filtered sequences were mapped to the reference *A. thaliana* genome using HISAT2. Subsequently, the aligned read files were processed by String Tie. After the reads were assembled into transcripts, their abundance was estimated and normalized using the numbers of reads Fragments Per Kilobase of transcript per Million fragments mapped reads (FPKM).

The differential expression analysis was performed using DEseq2. The threshold for differentially expressed genes (DEGs) was set as false discovery rates (FDR) ≤ 0.05 and fold change (FC) ≥ 1.5. Volcano plots were created based on these DEGs. To elucidate the particular DEGs’ functional characteristics, all DEGs were compared using the Kyoto Encyclopedia of Genes and Genomes (KEGG) reference database of pathway networks or integration by BLASTx. For the KEGG enrichment analysis, pathways with *p* ≤ 0.05 were considered significantly enriched. For heatmap analysis, the key genes enriched into pathways and with the similar expression pattern were targeted and selected to perform expression profiles analysis.

### 2.4. Confirmation of the RNA-Seq Data by Real Time Quantitative PCR (RT-qPCR)

The key genes involved in important steps of carbon fixation, including rubisco activase (*RCA*), glyceraldehyde 3-phosphate dehydrogenase subunit A (*GAPA*), fructose 1,6-bisphosphate aldolase (*FBA*), and sedoheptulose-1, 7-bisphosphatase (*SBPASE*), were selected for expression confirmation in the Arabidopsis and rice experimental plants, based on the obtained RNA-Seq data. The primer pairs for each selected gene were designed automatically using Primer6 (Appendix A). The RT-qPCR reactions were performed on Biorad (CFX 96) with Takara TB Green Kit (RR420Q) (TaKaRa, Dalian, China). The reaction volume was 25.0 μL, containing 12.5 μL TB Green Premix Ex Taq (Tli RNaseH Plus), 2.0 μL cDNA, 1.0 μL forward primer and 1.0 μL reverse primer (10 μM), and 8.5 μL ddH_2_O, following the manufactory protocols. All reactions were repeated three times. The RT-qPCR procedure was determined as 95 °C for 30 s and 40 cycles of 95 °C for 5 s, 60 °C for 30 s, and 72 °C for 30 s. The relative gene expression level was calculated with the 2^−△△Ct^ method [29] using ubiquitin 5 (*UBQ5*) [25] and *ACTIN* [30] as the reference gene in Arabidopsis and rice, respectively.

### 2.5. Statistical Analysis

The independent *t* test [30] was conducted to compare differences in expression of the *EPSPS* (trans)gene, intercellular CO_2_ concentration (Ci), expression of DEGs, expression of the four carbon-fixation related genes, and photosynthetic ratios between GE plants and their corresponding non-GE counterparts.

## 3. Results

### 3.1. Increases in EPSPS Gene Expression and Carbon Consumption in EPSPS Tansgenic Arabidopsis and Rice Lineages

Our results, based on the RT-qPCR method, showed significant increases (*p* < 0.001) in the average expression level of the *EPSPS* transgene in Arabidopsis and rice lineages. The *EPSPS* expression level was ~100–150 folds’ higher in the GE Arabidopsis lineages overexpressing the *EPSPS* gene (Figure 1, A1 and A2, black columns) than their corresponding non-GE lineages with normal *EPSPS* expression (Figure 1, A1 and A2, white columns). Similarly, the expression level of *EPSPS* gene was significantly higher (*p* < 0.001) in the GE rice lineage with ~50 fold higher gene expression than its corresponding normal non-GE lineage (Figure 1, W).

The concentration of CO_2_ in intercellular (Ci), as an estimate of the carbon consumption, was directly measured in the experimental plants. Results from this study indicated a significantly lower level (*p* < 0.05) of remaining CO_2_ in the GE Arabidopsis and rice lineages with overexpression of the *EPSPS* gene (Figure 2, black columns) than that in their corresponding non-GE counterparts with normal *EPSPS* expression (Figure 2, white columns). The Ci decreased 3.2% to 3.7% in the GE Arabidopsis and rice lineages, compared with their corresponding non-GE counterparts with normal *EPSPS* expression. These results suggested that GE Arabidopsis and rice lineages consumed more CO_2_ than their corresponding non-GE counterparts in terms of the directly measured Ci values.

Further, we included another value (the Ca-Ci value) to indicate the amount of CO_2_ consumption during the photosynthetic assimilation for accurate estimate of the relationships between the level of increased exogenous CO_2_ consumption and expression of the *EPSPS* gene in the GE and non-GE Arabidopsis (Figure 3a) and rice plants (Figure 3b). The obtained results showed that overexpression of the *EPSPS* gene consumed 14–20% greater CO_2_ in the GE Arabidopsis and rice lineages than their corresponding non-GE counterparts with normal *EPSPS* expression. There was a significantly positive correlation (*R^2^* = 0.70–0.72, *p* < 0.05) between the levels of CO_2_ consumption (as represented by the Ca-Ci values) and *EPSPS* gene expression in the GE and non-GE Arabidopsis and rice lineages (Figure 3a,b). These results indicated that a greater amount of CO_2_ was consumed within a given time to meet the demands for carboxylation in the photosynthesis of the *EPSPS* GE plants having greater *EPSPS* gene expression.

### 3.2. Pathways with Differential Gene Expression Caused by Overexpressing the EPSPS Gene between GE and Non-GE Arabidopsis Lineages

To further investigate the underlying mechanism concerning why overexpression of the exogenous *EPSPS* gene increases the CO_2_ consumption, we analyzed the GE and non-GE Arabidopsis lineages—the most commonly used model plants through the analysis of RNA-seq. Our results, based on the assay of differentially expressed genes (DEGs) in the Arabidopsis plants, indicated the considerable impact of overexpressing the *EPSPS* gene on the expression pattern of the global genes, although considerable variation in gene expression was observed between the two Arabidopsis lineages (Figure 4a,b; Appendix A). As shown in Figure 4a, 1113 DEGs were detected in the A1 Arabidopsis GE and non-GE lineages (Figure 4a). Of these DEGs, 656 genes displayed down-regulation and 457 showed upregulation. However, different from the A1 Arabidopsis lineage, 3766 DEGs were detected in the A2 Arabidopsis GE and non-GE lineages (Figure 4b). Of these DEGs, 1857 genes displayed down-regulation and 1909 showed upregulation.

Results based on further analyses of the Kyoto Encyclopedia of Genes and Genomes (KEGG) enrichment indicated that DEGs associated with overexpression of the *EPSPS* (trans)gene were categorized into different biological pathways of two Arabidopsis lineages (Figure 5). Unexpectedly, 271 (~31%) and 1281 (~39%) of the total DEGs were annotated by KEGG and were enriched in a total of 97 and 121 important pathways of the A1 and A2 Arabidopsis lineages, respectively (Figure 5a,b; Appendix A). Among these top 25 selected and displayed pathways, the porphyrin and chlorophyll metabolism, carbon fixation in photosynthetic organisms, phenylalanine, tyrosine and tryptophan biosynthesis, tryptophan metabolism, flavone and flavonol biosynthesis, and phenylpropanoid biosynthesis were included, which all showed a significant enrichment. These pathways were determined to be closely associated with overexpression of the *EPSPS* gene and carbon fixation. The above results might provide some insights into the mechanisms to explore the relationships between increased CO_2_ consumption and overexpression of the *EPSPS* (trans)gene.

Results from the KEGG enhancement further indicated similar expression profiles of the critical genes (Figure 6) that were supposedly involved in the pathways closely associated with *EPSPS* gene overexpression or carbon fixation. Interestingly, some pivotal carbon fixation genes (i.e., *GAPA*, *GAPB*, *EMB3119*, and *FBA1*), particularly those involved in the photosynthetic ratio limiting steps (i.e., *RCA* and *SBPASE*), showed upregulated expression although with considerable variation in their expression levels between the A1 and A2 Arabidopsis lineages (Figure 6). The expression level of most chlorophyll biosynthesis genes (e.g., *HEMA1*, *GUN5*, *PCB2*, and *ATG4*) was significantly increased with overexpression of the *EPSPS* gene in this study (Figure 6). In addition, most genes (e.g., *ASA1*, *TSB2*, *CM1*, *PAT*, and *PD1*) that hypothetically participated in the biosynthesis of aromatic amino acids (i.e., phenylalanine, tyrosine, and tryptophan) and genes (e.g., *CAD4*, *PAL2*, *CYP83B1*, and *UGT78D1*) that were associated with metabolisms also showed significantly higher expression (Figure 6). Interestingly, genes that were supposed to be involved in the translocation (*PPT2*) and the last-three-steps in the biosynthesis of phosphoenolpyruvate (PEP)—one of the natural substrates of EPSPS enzyme–were also detected to be upregulated in terms of their expression (Figure 6).

### 3.3. Increases in Expression of the Key Carbon-Fixation Genes and Photosynthetic Ratios of GE Arabidopsis and Rice Lineages

Four key carbon-fixation genes were selected to confirm their actual expression level in the Arabidopsis and rice experimental plants. Consistently, the expression level of the four key genes that showed generally high expression in the RNA-seq analysis also displayed significant upregulation (*p* < 0.05), both in the Arabidopsis and rice plants using the RT-qPCR method (Figure 7). Expression of the *RCA* gene, which activated Rubisco responsible for catalyzing the primary step of CO_2_ fixation, was significantly increased in the GE Arabidopsis (A1 by 1.9 folds and A2 by 1.8 folds) and in the GE rice lineage (W by 1.7 folds), compared with that in their corresponding non-GE counterparts (Figure 7a). Expression of the *GAPA* gene that is associated with the reduction action step, the main product exit point of Calvin–Benson Cycle, also showed significant upregulation in the GE Arabidopsis lineages (A1 by 2.4 folds and A2 by 2.2 folds) and in the GE rice lineage (W by 4.6 folds), compared with that in their corresponding non-GE counterparts (Figure 7b). Similarly, the two genes that are involved in RuBP regeneration (e.g., the *FBA* and *SBPASE* genes), affecting the maximum CO_2_ assimilation rates, showed upregulation in the GE Arabidopsis and rice lineages for different folds (Figure 7c,d). These results all indicated that overexpressing the *EPSPS* gene could elevate expression of the carbon-fixation related genes, and consequently increased the CO_2_ consumption of the GE plants.

To confirm whether the overexpression of the *EPSPS* gene could enhance photosynthesis, we measured photosynthetic ratios of the experimental plants using the LI-6800 portable photosynthetic system. Our results indicated significantly greater photosynthetic ratios (*p* < 0.01) in the GE Arabidopsis and rice lineages than those in their corresponding non-GE counterparts (Figure 8). The photosynthetic ratios were ~30% (A1) and ~40% (A2) greater in the two GE Arabidopsis lineages than those in their corresponding non-GE counterparts (Figure 8, A1 and A2). Similarly, our results also showed significantly greater photosynthetic ratio (*p* < 0.001) in the GE rice lineage, by ~34%, than that in its corresponding non-GE counterpart (Figure 8, W). These results generated from both monocotyledonous Arabidopsis and dicotyledonous rice suggested that overexpressing the *EPSPS* gene might generally increase the photosynthesis of all plants.

## 4. Discussion

### 4.1. Overexpressing the EPSPS Transgene Substantially Increased CO_2_ Consumption in Plants

In this study, we found that the expression level of the *EPSPS* gene increased substantially in the fourth generation (T_4_) of genetically engineered (GE) Arabidopsis plants and GE rice/weedy rice hybrid lineages (F_4_). The *EPSPS* gene was cloned from cultivated rice [31] and driven by the constitutive promotors CaMV35S (for GE Arabidopsis) and Ubiquitin (for GE rice) [22,25]. This finding was generated based on the comparative analyses of average mRNA expression between the GE Arabidopsis/rice plants and their corresponding non-GE counterparts. The GE plants overexpressed the *EPSPS* gene, whereas the non-GE plants had normal *EPSPS* expression. Importantly, our experimental materials included in this study represented both dicotyledonous (Arabidopsis) and monocotyledonous (rice) plants. Therefore, we expect that the proper modification of the *EPSPS* gene(s) through GE technologies can substantially increase the expression level of this gene both in dicots and monocots. This prediction is supported by previous results published by different authors [22,23,25,26]. In addition, the detected expression level of the *EPSPS* gene(s) in the GE Arabidopsis and rice plants was largely variable. Therefore, it is possible to select transgenic events with substantially enhanced *EPSPS* overexpression in particular plants using more effective GE technologies.

Interestingly, we also found in this study that the intercellular CO_2_ concentration (Ci) was significantly reduced both in the GE Arabidopsis and rice plants with significantly greater *EPSPS* gene expression, although the measurement for Ci only lasted for about ten minutes. The intercellular CO_2_ concentration, as a critical parameter related to carbon fixation, was reported in a previous study where carbon fixation was determined by the carboxylation activity or activated amount of ribulose-1,5-bisphosphate carboxylase/oxygenase [32]. The published results supported our findings that reduced intercellular CO_2_ concentration is a critical determinant to measure carbon fixation. In addition, we also detected a significant positive correlation (R^2^ > 0.70, *p* < 0.05) between the levels of atmospheric CO_2_ consumption (Ca-Ci) and *EPSPS* gene expression when the parameter Ci was converted into Ca-Ci. Altogether, these findings suggest that overexpression of the *EPSPS* gene in our experimental plants can considerably increase the consumption or fixation of atmospheric CO_2_. This conclusion addressed our first question concerning overexpression of the *EPSPS* transgene and fixation of CO_2_ in plants. In our study, ~20% greater amount of CO_2_ was consumed by the GE experimental materials than their corresponding non-GE counterparts. Based on these results, we can make a prediction that the appropriate application of overexpression of the *EPSPS* gene(s) in plants through GE technologies can considerably increase fixation of CO_2_. Statistical data showed that one third of CO_2_ is produced by anthropogenic activities [14,15]. Therefore, our findings provide a new opportunity for global carbon neutrality through genetically engineered plants.

### 4.2. Possible Mechanisms for Increased CO_2_ Consumption in Genetically Engineered Plants Overexpressing EPSPS Gene

Our results, based on the transcriptomic analyses, evidently demonstrated that overexpressing the *EPSPS* gene had a substantial impact on the expression level of many genes in different biological pathways. This finding is based on the comparison of differentially expressed genes (DEGs) between GE and non-GE Arabidopsis plants. Particularly, our results from KEGG (Kyoto Encyclopedia of Genes and Genomes) analyses indicated significant gene enrichment in the carbon-fixation related pathways. All the carbon-fixation related pathways enriched in this study are found in the down-streams of the shikimate pathway, including the phenylalanine, tyrosine, and tryptophan biosynthesis pathways, and the phenylpropanoid biosynthesis pathway. The above results indicate that more carbon from the atmosphere can be fixed due to overexpression of the *EPSPS* gene(s). Obviously, overexpression of the *EPSPS* gene can trigger upregulation of many other genes in the carbon-fixation related pathways.

This explanation is supported by many previously published results in which overexpression of the *EPSPS* gene significantly increased the production of aromatic amino acids [22,23], lignin [33], and biomasses [22,23,25] of the GE Arabidopsis and rice plants. All the produced metabolic matters mentioned above are closely associated with the carbon-fixation related pathways [27,28]. As a matter of fact, metabolites of phenylpropanoid and aromatic amino acids, such as phenylalanine, tyrosine, and tryptophan, are synthesized in the carbon-fixation related pathways, accounting for ~20% of total carbon fixed in plants [27,28]. In addition, our results also indicated upregulated expression of the PPT2 gene that is reported to be responsible for translocating EPSPS’s substrate PEP—the carbon skeleton of the shikimate-pathway derived metabolites [34,35]. These results again suggest more carbon being fixed in plants by overexpressing the *EPSPS* gene. Published results also indicated that increased expression of the PPT gene can stimulate carbon fixation by increasing the content of aromatic amino acids, the number of fruits, and biomasses of plants [36]. Therefore, our conclusion in this study, that overexpression of the *EPSPS* gene(s) can fix more carbon by enhancing expression of many genes, gains supports from previous published results.

Accordingly, we examined the expression of four important carbon-fixation related genes, *RCA*, *GAPA*, *FBA*, and *SBPASE*, with a consistent upregulating pattern in the two examined GE Arabidopsis lineages based on our RNA-seq analysis. In addition to our results, published reports also indicated that these genes played critical roles in the three phases of the carbon-fixation pathway in most organisms with photosynthesis [37]. The four genes were found to be significantly upregulated in Arabidopsis and rice plants overexpressing the *EPSPS* gene. This finding confirms the consistency of the upregulation patterns concerning carbon-fixation related genes between the RT-qPCR and RNA-seq analyses from our experiment. We consider that the increases in CO_2_ fixation in the GE Arabidopsis and rice plants is likely due to upregulation of the key genes in the carbon-fixation related pathways because of the overexpression of the *EPSPS* gene. This conclusion is based on our findings about the upregulation of carbon-fixation related genes.

In fact, the expression changes of the four genes (*RCA*, *GAPA*, *FBA*, and *SBPASE*) have downstream effects on plants’ physiology or metabolism, including photosynthesis, biomass, and plant growth. The previous studies also indicated that the four genes were the key to the three phases of carbon fixation, namely, carbon fixation (*RCA*), reduction reaction (*GAPA*), and ribulose 1,5-bisphosphate regeneration (*FBA* and *SBPASE*) [37]. First, the *RCA* gene is essential for the activation and maintenance of catalytic activity of rubisco that catalyzes the first step of carbon fixation [38,39,40]. Overproduction of *RCA* can enhance the carbon fixation in Arabidopsis and rice [41,42]. Second, the *GAPA* gene encodes the subunits GapA of GAPDH involving in the reduction action of carbon fixation. Overexpression of GAPDH can increase the ratios of carbon fixation in rice [43]. Third, the *FBA* gene is associated with photosynthetic carbon flux [44,45], and increased *FBA* in plastids can enhance carbon fixation of tobacco plants [46]. The *SBPASE* gene is essential for maintaining the balance between the carbon required for the regeneration of ribulose 1,5-bisphosphate and required for the biosynthesis of different polysaccharides [47]. In addition, it was found in wheat and Arabidopsis that increased *SBPASE* activities could also improve carbon fixation [48,49]. All these results and findings support our conclusion that upregulation of the key genes in the carbon-fixation related pathways through overexpressing the *EPSPS* gene enhances carbon fixation.

We also found increased photosynthetic ratios and critical genes that are involved in chlorophyll biosynthesis [50] in the GE Arabidopsis and rice plants overexpressing the *EPSPS* gene. Previous studies reported that enhanced chlorophyll biosynthesis is closely related to photosynthesis [51,52]. All these findings provide evidence for the explanation of increased photosynthesis in the GE rice [22] and Arabidopsis in our study. Given that increased photosynthesis is also associated with increased CO_2_ assimilation [53], our findings regarding overexpression of the *EPSPS* gene that enhances carbon fixation provide indirect evidence of increased photosynthesis from another perspective. In summary, we propose that increased CO_2_ fixation caused by overexpression of the *EPSPS* gene is most likely owing to the upregulation of the key genes in the carbon-fixation related pathways. Increased photosynthetic ratios provide additional evidence for increased carbon fixation in plants. These findings facilitate our understanding about the relationships between overexpression of the *EPSPS* gene(s) and carbon fixation in plants. These findings can also provide theoretical foundation for regulating carbon fixation and crop production through genetic engineering of plant species with enhanced expression of the endogenous or exogenous *EPSPS* gene(s).

### 4.3. Implications of Overexpressing EPSPS in Genetically Engineered Plants for Carbon Neutrality

Our results clearly indicated significant variation in CO_2_ fixation among different GE Arabidopsis and rice plants coped with diverse levels of *EPSPS* gene expression. This finding provides great opportunities for us to select a certain transgenic event that has an extremely high level of *EPSPS* gene expression as a candidate for ideal CO_2_ fixation in the plant breeding programs. There are many methods available that can regulate the expression level of GE plants, such as the use of different promotors with variable driving efficiencies for the same genes [54] and the selection of exogenous and endogenous genes from different sources for genetic engineering [22,25,55,56]. We also found a significantly positive correlation between the levels of carbon fixation, as represented by Ca-Ci that showed considerable variation, and expression of the *EPSPS* gene in this study. These results suggest that the expression level of the *EPSPS* gene can determine the efficiency of carbon fixation in transgenic events of different GE plants of the same species or hybrid lineages. If the hypothesis is true, we can deliberately apply different GE technologies to increase the expression level of the *EPSPS* gene(s) in different plant species to achieve the most ideal carbon fixation efficiencies. Furthermore, we found in our transcriptomic analyses that overexpressing the *EPSPS* gene enhanced the expression level of the key genes in carbon-fixation related pathways, resulting in increased carbon fixation. These findings also suggest the possibility of an alternative strategy by selecting other proper genes, such as the *RCA* and *SBPASE* genes, in the carbon-fixation related pathways to enhance their (over)expressing level through GE technologies for achieving the direct increases in carbon fixation.

Accordingly, we can develop better strategies to effectively reduce the concentration of atmospheric CO_2_ to achieve successful carbon neutrality through carbon fixation, applying the biotechnology of genetic engineering, particularly in the cropland, even in artificial woodland and artificial grassland. Noticeably, the *EPSPS* genes are found in all plant species, responsible for the biosynthesis of important secondary metabolites, such as lignin and auxin [28]. Theoretically, it is possible to increase expression of the *EPSPS* genes in all plant species for enhanced CO_2_ fixation by genetic engineering. Taking the above facts into consideration, we believe that more atmospheric CO_2_ can be assimilated or consumed by GE plants that overexpress the *EPSPS* genes. Therefore, different technologies can be selected and applied to modify plant species that are suitable to grow in different terrestrial ecosystems, such as in croplands and artificial woodlands. Ideal strategies that combine the proper genetic engineering technologies to modify the *EPSPS* genes with the properly selected plant species to fix carbon (CO_2_) efficiently in different areas will offer a great potential for human to meet the challenges of the global climate change and to ensure carbon neutrality, which may offset a large portion of anthropogenic CO_2_ emissions.

Cropland that has a great potential to develop GE plants covers ~10% of the Earth’s land surface [57], with the estimated coverage of 1.24 billion hectares in 2019 [58]. Given that GE crops are extensively grown in the world, with more than 190 million hectares in 2019 [59], it is most likely to develop GE crops overexpressing the *EPSPS* genes. For example, crop rice (*Oryza sativa*) is cultivated worldwide, accounting for ~11% of the global cropland with an estimated cultivation area of 165.5 million hectares [60]. As an important crop that provides staple food for nearly one half of the global population [61], this plant species alone already provides significant opportunities for reducing atmospheric CO_2_ by genetic engineering of its endogenous, as well as exogenous, *EPSPS* genes. The alternative example is to engineer tree species, because the global coverage of forests is ~4.06 billion hectares—approximately 30% of the total terrestrial area [15]. Usually, tree species occurring in forests have a long growth period, which can promote sustainable carbon fixation during their lifespans. It is reported that the global coverage of artificial woodlands increased by 4.1 million hectares per year between 2000~2020 and the current coverage is ~293 million hectares, as estimated in 2023 [62]. Therefore, tree species, particularly the artificial woodland species, provide ideal plants for enhanced carbon fixation by genetic engineering. In addition, plant species occurring in the grasslands can also be used to meet the objectives, as grasslands cover ~3.5 billion hectares, accounting for ~26% of the ice-free land on Earth [63].

Therefore, the possibilities of applying plant species in different areas in combination with advanced biotechnologies for genetic engineering allow human being to manage CO_2_ fixation from multiple dimensions to achieve global carbon neutrality. However, it is necessary to point out the potential challenges and limitations regarding the application of GE plants. Currently, GE plants cannot be freely grown in the world. According to the biosafety regulations, all GE plants that are intended to be released to the environment for large-scale application should be subjected to risk assessment [30,64,65,66]. Only those GE plants that receive a biosafety certificate are permitted to grow in a given area. Therefore, the potential challenges and limitations of applying GE plants overexpressing the *EPSPS* genes associated with biosafety issues need to be considered and addressed properly.

## 5. Conclusions

In this study, we found substantially reduced intercellular carbon dioxide (CO_2_) concentration (Ci) in genetically engineered (GE) Arabidopsis (dicot) and rice (monocot) plants that overexpressed the rice *EPSPS* gene. Further analyses indicated a positive correlation between the levels of atmospheric CO_2_ consumption (Ca-Ci) and *EPSPS* gene expression in these Arabidopsis and rice plants. The possible reason for substantially increased atmospheric CO_2_ consumption by overexpressing the *EPSPS* gene is upregulation of many key genes in biological pathways, particularly those in the carbon-fixation related pathways. For example, expression of the carbon-fixation associated genes such as *RCA, GAPA, FBA*, and *SBPASE* showed significant increases. Given that our obtained results are based on the dicot (Arabidopsis) and monocot (rice) plants, overexpression of *EPSPS* gene(s) promoting increased CO_2_ consumption may represent more plant species in nature. Altogether, findings from this study have a great implication to design a novel strategy not only for the global carbon neutrality as the carbon sink, but also for increases in food production due to increases in photosynthetic ratios. In other words, overexpression of endogenous or exogenous *EPSPS* genes in plants through GE biotechnologies can promote a higher level of consumption of atmospheric CO_2_ in different ecosystems, which is important for global carbon neutrality.

## Figures and Tables

**Figure 1 biology-13-00025-f001:**
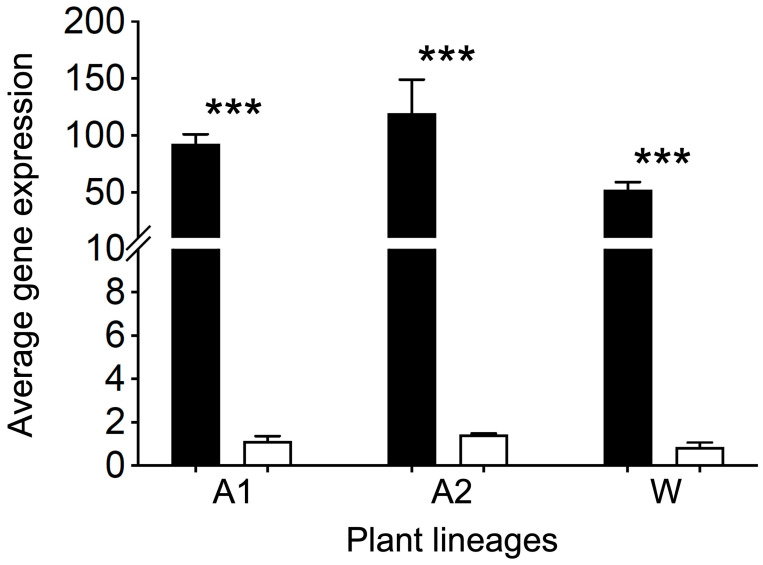
Comparison of the average expression (relative mRNA) level of the rice *EPSPS* (trans)gene between the GE lineages (black columns) overexpressing the *EPSPS* gene and their corresponding non-GE lineages (white columns), based on the independent *t* test [30]. A1 and A2 represent the Arabidopsis lineages; W represents rice lineage. Bars indicate standard errors; *** *p* < 0.001.

**Figure 2 biology-13-00025-f002:**
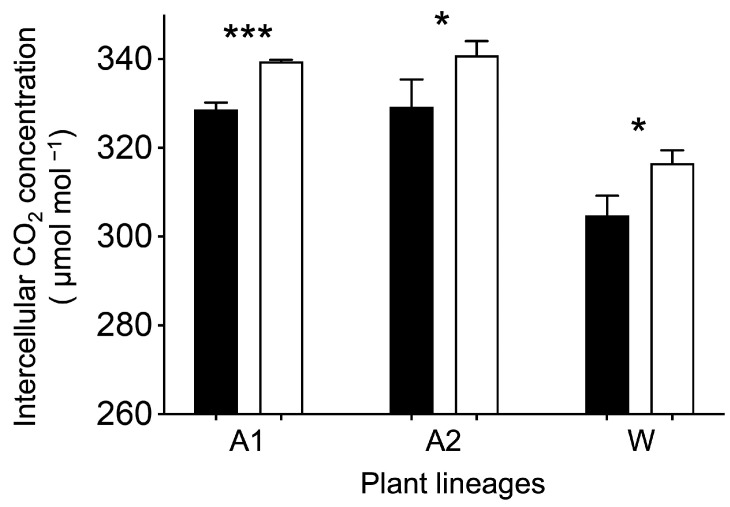
Comparison of average intercellular carbon dioxide (CO_2_) concentration between the GE lineages (black columns) overexpressing the *EPSPS* gene and their corresponding non-GE lineages (white columns), based on the independent *t* test [30]. A1 and A2 represent Arabidopsis lineages; W represents rice lineage. Bars represent standard errors; * *p* < 0.05, *** *p* < 0.001.

**Figure 3 biology-13-00025-f003:**
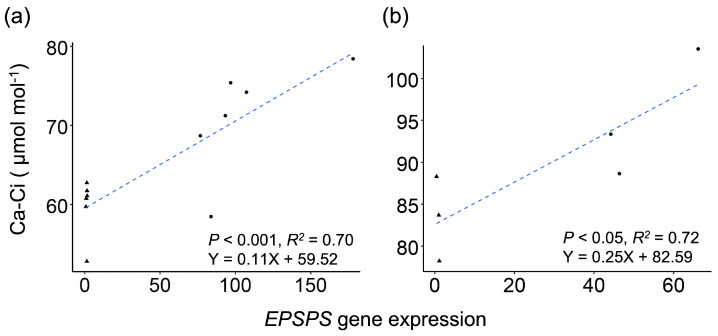
Correlation between the calculated consumption level (Ca-Ci) of carbon dioxide (CO_2_) and *EPSPS* (trans)gene expression in the GE (black dots) and non-GE (black triangles) Arabidopsis (**a**) and rice (**b**) plants. The Ca-Ci value is calculated as the differences between measured air CO_2_ concentrations (Ca) and intercellular CO_2_ concentrations (Ci). Dashed lines indicate linear regression.

**Figure 4 biology-13-00025-f004:**
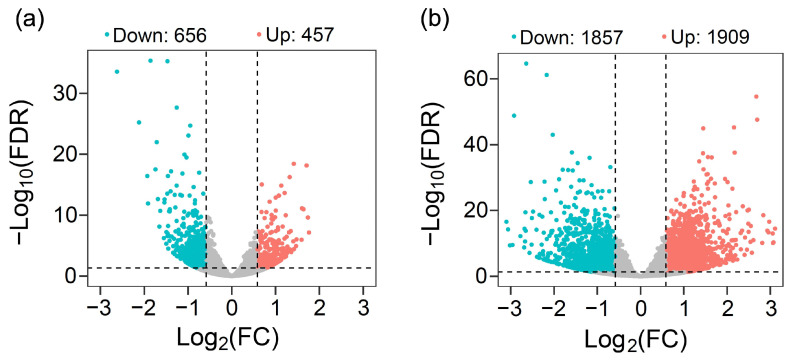
Significantly up-regulated (red dots) and down-regulated genes (light-blue dots) indicated by the volcano plots based on the differentially expressed genes (DEGs) between the GE plants overexpressing the *EPSPS* gene and their corresponding non-GE Arabidopsis plants (A1, (**a**) and A2, (**b**)). FDR represents the false discovery rates with significances; FC represents the fold changes, indicating the expression fold changes of genes between the GE plants overexpressing the *EPSPS* gene and their corresponding non-GE plants; Grey dots indicate genes with no significant expression changes. Dashed lines indicate the significance threshold of FDR ≤ 0.05 (horizontal) and the fold change threshold of FC ≥ 1.5 (longitudinal), respectively.

**Figure 5 biology-13-00025-f005:**
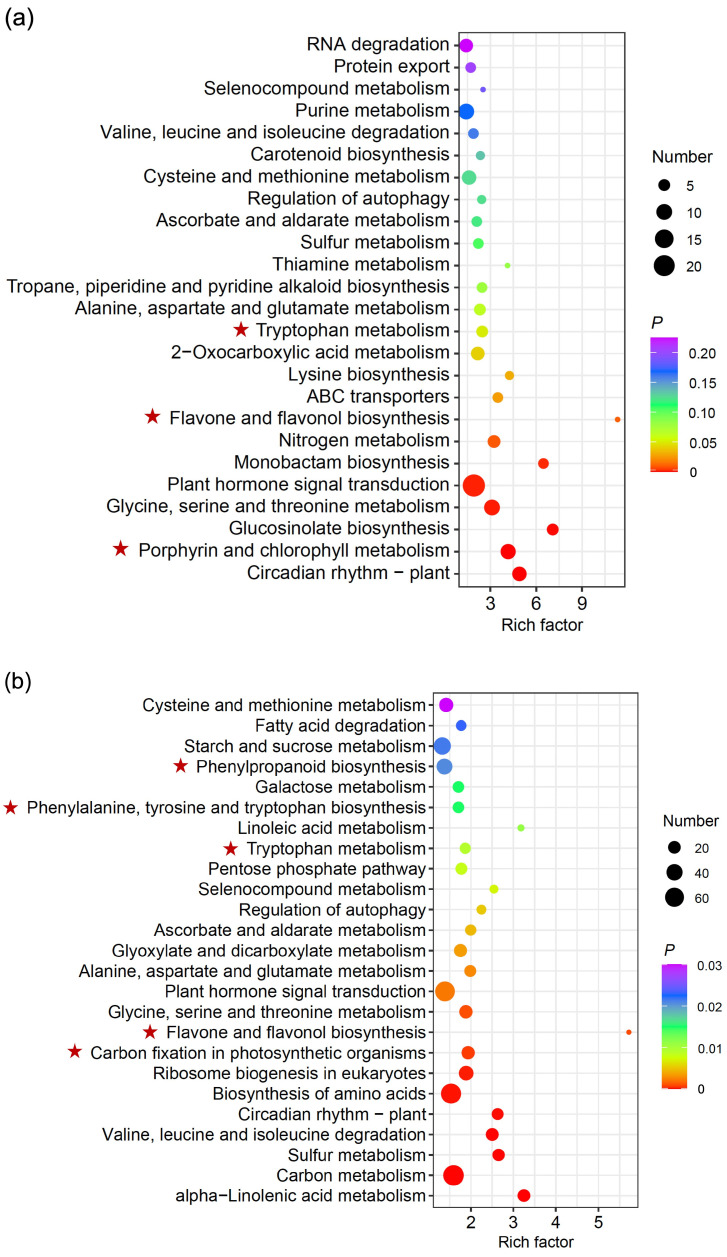
Biological pathways with gene expression changes caused by overexpression of the *EPSPS* gene based on the Kyoto Encyclopedia of Genes and Genomes (KEGG) enrichment in the GE plants overexpressing the *EPSPS* gene and their corresponding non-GE Arabidopsis plants (A1, (**a**) and A2, (**b**)). Red stars represent pathways closely related to carbon fixation. The rich factor indicates the significance of the enrichment level; *P* represents the reliability of the significance of the enrichment level.

**Figure 6 biology-13-00025-f006:**
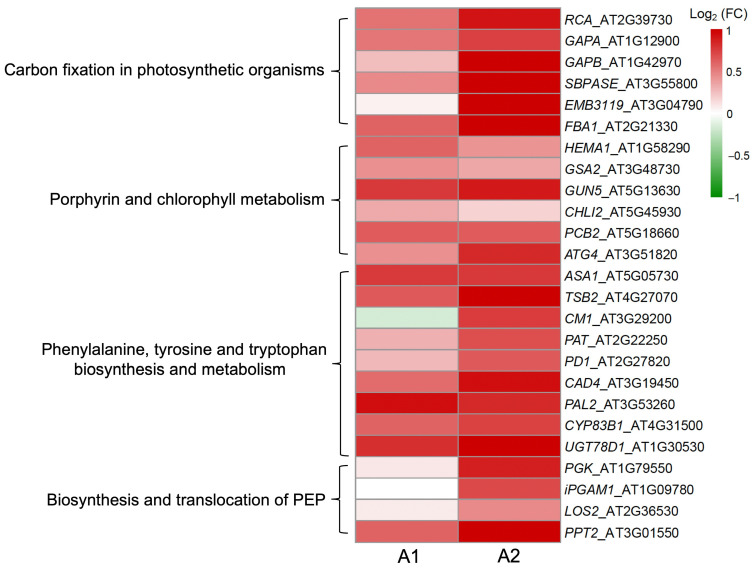
Variation in the expression profiles of the target genes in the selected pathways closely associated with *EPSPS* function and carbon fixation between the A1 and A2 Arabidopsis lineages based on the Heatmap. FC indicates the expression fold change of genes between the GE lineages overexpressing the *EPSPS* gene and their corresponding non-GE lineages.

**Figure 7 biology-13-00025-f007:**
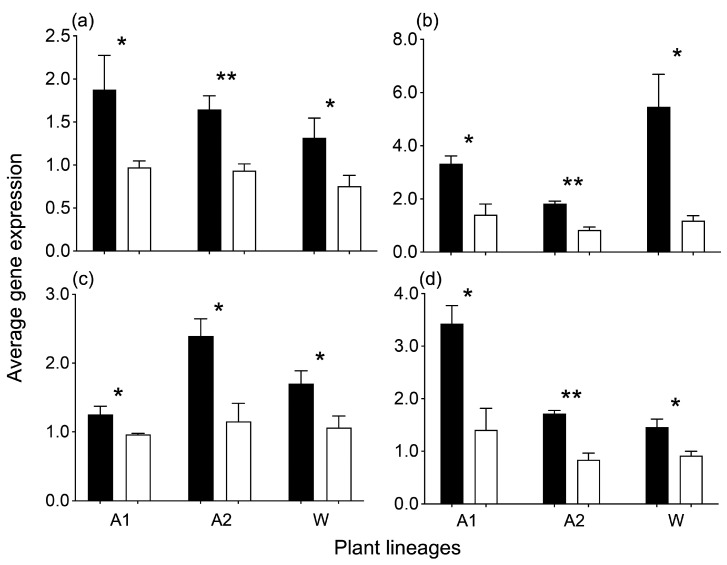
Comparison of expression levels of the four carbon-fixation related genes using the RT-qPCR method between GE (overexpressing the *EPSPS* gene, black columns) and non-GE (white columns) Arabidopsis (A1 and A2) and rice (W) lineages, based on the independent *t* test [30]. Bars represent standard errors; (**a**–**d**) indicate the RCA, GAPA, FBA, and SBPASE genes, respectively. * *p* < 0.05, ** *p* < 0.01.

**Figure 8 biology-13-00025-f008:**
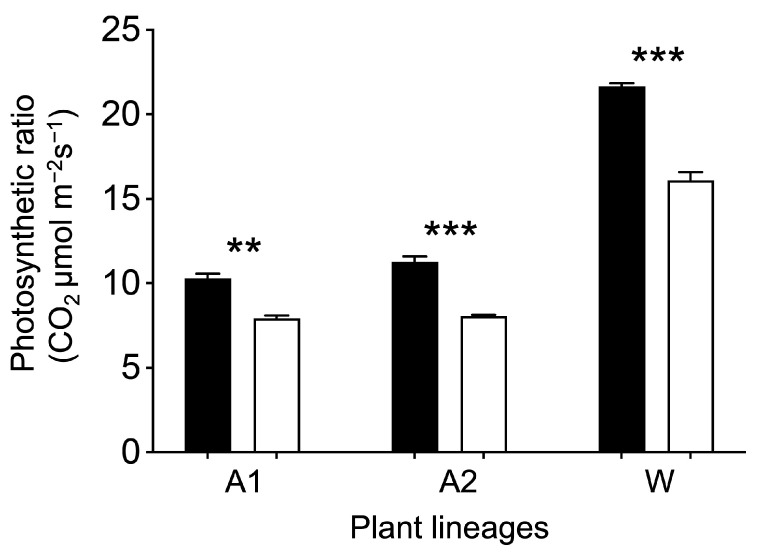
Comparison of the average photosynthetic ratios between the GE (overexpressing the *EPSPS* gene, black columns) and non-GE (white columns) Arabidopsis (A1 and A2) and rice (W) lineages, based on the independent *t* test [30]. Bars represent standard errors; ** *p* < 0.01, *** *p* < 0.001.

## Data Availability

All data are included in the article and Appendix A.

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
