# Peer review of "Reduced Carbon Dioxide by Overexpressing EPSPS Transgene in Arabidopsis and Rice: Implications in Carbon Neutrality through Genetically Engineered Plants"

_biology, 2023, doi:10.3390/biology13010025_

Round 1

Reviewer 1 Report

Comments and Suggestions for Authors

Dear Author,

I hope this letter finds you well. I would like to praise you on the thorough research presented in your manuscript titled "Reduced Carbon Dioxide by Overexpressing EPSPS Transgene in Arabidopsis and Rice: Implications in Carbon Neutrality through Genetically Engineered Plants."

I have carefully reviewed the manuscript and found it to be well-structured and supported by robust data. However, I believe that a few minor corrections and suggestions for improvement could enhance the clarity and impact of your findings. Kindly take into account the following points:

1.    Can you provide more details on the methodology used to measure atmospheric CO2 consumption in Arabidopsis and rice plants with overexpressed EPSPS genes?

2.    What specific environmental conditions were maintained during the experiments, and how were they controlled to ensure accurate measurements?

3.    Were there any specific challenges or variations observed in the RNA-seq data that may have influenced the analysis of DEGs?

4.    Regarding the upregulation of carbon-fixation-related genes (RCA, GAPA, FBA, and SBPASE), what downstream effects on plant physiology or metabolism are anticipated due to their increased expression?

5.    The study mentions variability in EPSPS gene expression levels. Could you provide insights into the potential factors influencing this variability, and how it might impact the overall findings?

6.    How might the findings of this study be practically applied or translated into agricultural practices for enhanced carbon fixation? Are there potential limitations or challenges that need to be considered?

Comments on the Quality of English Language

1.    In section 4.3, "diverse level" should be "diverse levels" [Line 440]

2.    Although the manuscript is well written, some complex sentences need to be addressed to enhance the clarity and readability of the paper. I recommend carefully reviewing the manuscript to ensure accurate grammar and punctuation, as well as considering breaking down some of the longer sentences to improve understanding. Rest all good.

Author Response

POINT-BY-POINT RESPONSES TO REVIEWER #1

General comment: I hope this letter finds you well. I would like to praise you on the thorough research presented in your manuscript titled "Reduced Carbon Dioxide by Overexpressing EPSPS Transgene in Arabidopsis and Rice: Implications in Carbon Neutrality through Genetically Engineered Plants."

I have carefully reviewed the manuscript and found it to be well-structured and supported by robust data. However, I believe that a few minor corrections and suggestions for improvement could enhance the clarity and impact of your findings. Kindly take into account the following points:

Response: We indeed appreciate the constructive evaluation of our manuscript by the Reviewer #1, and we shall take all the Reviewer’s points and suggestions into consideration when we revise our manuscript.

Comment-1. Can you provide more details on the methodology used to measure atmospheric CO2 consumption in Arabidopsis and rice plants with overexpressed EPSPS genes?

Response: Thanks for the comment. The atmospheric CO2 consumption (Ca-Ci) was calculated based on the differences between measured air CO2 concentrations (Ca) and intercellular CO2 concentrations (Ci). Accordingly, we have made necessary modification concerning how to measure the atmospheric CO2 consumption in Material and Methods (2.2. Gas-exchange measurements) to make the methodology clearer in the revised manuscript.

Comment-2. What specific environmental conditions were maintained during the experiments, and how were they controlled to ensure accurate measurements?

Response: We appreciate the comment and have added more detailed information about the specific environmental conditions in Material and Methods (2.2. Gas-exchange measurements) in the revised manuscript.

Comment-3. Were there any specific challenges or variations observed in the RNA-seq data that may have influenced the analysis of DEGs?

Response: Thanks for the concern. In fact, there is no significant variation between the biological replicates of examined lineages and the expression pattern of the examined carbon-fixation related DEGs is similar.

Comment-4. Regarding the upregulation of carbon-fixation-related genes (RCA, GAPA, FBA, and SBPASE), what downstream effects on plant physiology or metabolism are anticipated due to their increased expression?

Response: We appreciate this comment. We expected that upregulation of carbon-fixation-related genes (RCA, GAPA, FBA, and SBPASE) may have downstream effect on physiology or metabolism, including photosynthesis, biomass, and plant growth as we discussed in our manuscript. These results are consistent with the published results in our references [41-43, 46, 48, 49] already cited in our study. However, we have added a few sentences to explain this in the revised manuscript.

Comment-5. The study mentions variability in EPSPS gene expression levels. Could you provide insights into the potential factors influencing this variability, and how it might impact the overall findings?

Response: Thanks for pointing out variability in EPSPS gene expression levels. In fact, we expected variation of EPSPS gene expression among different transgenic events, which is most likely due to insertion site differences of the EPSPS gene in plant genomes. Variation is commonly found in different transgenic events of plants as in many published reports. For example, our study also showed a significantly positive correlation between the level of CO2 consumption and EPSPS gene expression in the GE plants (as shown in Figure 3). Therefore, variation in EPSPS gene expression does not impact our main findings about the CO2 fixation in this study. Importantly, variation among different transgenic events provides opportunities for us to select more effective events for implications in practices as we already discussed in the manuscript.

Comment-6. How might the findings of this study be practically applied or translated into agricultural practices for enhanced carbon fixation? Are there potential limitations or challenges that need to be considered?

Response: Thanks for the concern. Yes, there are limitations or challenges of using GE plants for enhanced carbon fixation agriculture, because GE plants are not permitted to grow anywhere in the world. The commercial application of GE plants should be regulated by the biosafety assessment. Therefore, we have added some sentences concerning the limitations or challenges of using GE plants for application in agricultural practices in Discussion of the revised manuscript.

Comments on the Quality of English Language

Comment-7. In section 4.3, "diverse level" should be "diverse levels" [Line 440]

Response: Thanks for the comment. We have changed this phrase “diverse level” to “diverse levels” in the revised manuscript.

Comment-8. Although the manuscript is well written, some complex sentences need to be addressed to enhance the clarity and readability of the paper. I recommend carefully reviewing the manuscript to ensure accurate grammar and punctuation, as well as considering breaking down some of the longer sentences to improve understanding. Rest all good.

Response: We indeed appreciate the suggestions. We have carefully checked and revised the English language of this manuscript to make the manuscript easier to be understood.

Reviewer 2 Report

Comments and Suggestions for Authors

This manuscript reports interesting research results in which the significant reduction of carbon dioxide (CO2) is associated with by overexpression of the exogenous (in Arabidopsis) and endogenous (in rice) EPSPS gene in different plant species produced by using the genetic engineering biotechnology. In practices, many crop species containing the genetically engineered (GE) EPSPS gene have been widely applied to tolerate the glyphosate herbicide for the management of agricultural weeds so far. Interestingly, the authors found another function of the overexpressed EPSPS genes to reduce carbon dioxide in GE plants, in addition to increase photosynthesis, and this finding opens another door for the potential implications in carbon neutrality, in addition to increase crop production by increased photosynthesis, through genetic engineering of plant species. Furthermore, the authors also found that overexpression of the EPSPS gene triggered up-regulation of many other genes in biological pathway, especially those in the carbon-fixation related pathways. The series results together are sufficiently supportive to demonstrate that CO2 consumption is enhanced in the GE Arabidopsis and rice plants by overexpressing the EPSPS gene. Therefore, results generated form this manuscript is novel and very useful. In summary, this manuscript provides novel knowledges in the related research field and potential strategy for global CO2 reduction. I recommend the publication of the manuscript in Biology.

I just have one concerned question.

1. Since the EPSPS gene is present in all plant species, can the level of expression of this gene be used to estimate the level of CO2 consumption in a given ecosystem?

Author Response

POINT-BY-POINT RESPONSES TO REVIEWER #2

General comment. This manuscript reports interesting research results in which the significant reduction of carbon dioxide (CO2) is associated with by overexpression of the exogenous (in Arabidopsis) and endogenous (in rice) EPSPS gene in different plant species produced by using the genetic engineering biotechnology. In practices, many crop species containing the genetically engineered (GE) EPSPS gene have been widely applied to tolerate the glyphosate herbicide for the management of agricultural weeds so far. Interestingly, the authors found another function of the overexpressed EPSPS genes to reduce carbon dioxide in GE plants, in addition to increase photosynthesis, and this finding opens another door for the potential implications in carbon neutrality, in addition to increase crop production by increased photosynthesis, through genetic engineering of plant species. Furthermore, the authors also found that overexpression of the EPSPS gene triggered up-regulation of many other genes in biological pathway, especially those in the carbon-fixation related pathways. The series results together are sufficiently supportive to demonstrate that CO2 consumption is enhanced in the GE Arabidopsis and rice plants by overexpressing the EPSPS gene. Therefore, results generated form this manuscript is novel and very useful. In summary, this manuscript provides novel knowledges in the related research field and potential strategy for global CO2 reduction. I recommend the publication of the manuscript in Biology.

I just have one concerned question.

Response: We indeed appreciate the positive comment on our manuscript by the Reviewer #2.

Comment-1. Since the EPSPS gene is present in all plant species, can the level of expression of this gene be used to estimate the level of CO2 consumption in a given ecosystem?

Response: Thanks for this comment. Results from this study showed a positive correlation between the level of carbon fixation and EPSPS gene expression. These results only suggest that the expression level of the EPSPS gene might be used to determine the efficiency of carbon fixation in different transgenic events of GE plants. If the suggested relationship is proven by more GE plant species/events, the level of gene expression might be used to estimate the level of CO2 consumption by the transgenic plants, but not to an ecosystem that is far more complicated.

Reviewer 3 Report

Comments and Suggestions for Authors

Materials and methods

I suggest that in Materials and methods you briefly state the basic data about the transgene, such as which promoter were used and which methods were used for the genetic modification of Arabidosis and which for rice!

Do you know which is the copy number of the EPSPS trangene in each lineages?

Are GE Arabidopsis and rice glyphosate tolerant?

In which condition the plants were grown? How many plants were used in this experiment?

 <=     ??   did you mean 

>=     ??   did you mean 

Discussion

… the increases in CO2 fixation in the GE Arabidopsis and rice plants observed in this experiment is likely due to upregulation of the key genes in the carbon-fixation related pathways, because of overexpression of the EPSPS gene.        Is there any explanation why is this happening? What is the mechanism for upregulation of all those genes?

The authors are overly optimistic and idealistic about the benefits of breeding EPSPS overexpressing crop and forest species, forgetting that GE plants are not allowed to be grown in much of the world, and the propagation of GE forest species would not be so simple.

Author Response

POINT-BY-POINT RESPONSES TO REVIEWER #3

Comment-1. Materials and methods: I suggest that in Materials and methods you briefly state the basic data about the transgene, such as which promoter were used and which methods were used for the genetic modification of Arabidosis and which for rice!

Response: Thanks for the comment and we have added more detailed information and references about the transgenic constructs in the Materials and Methods (2.1. Plant materials) in the revised manuscript.

Comment-2. Do you know which is the copy number of the EPSPS trangene in each lineages?

Response: Thank for this comment. We have added more detailed information and references about the copy of the EPSPS transgene in Materials and methods (2.1. Plant materials) in the revised manuscript.

Comment-3. Are GE Arabidopsis and rice glyphosate tolerant?

Response: Thanks for reminding this. Accordingly, we add more detailed information about the glyphosate tolerance of the transgene in in Materials and methods (2.1. Plant materials) in the revised manuscript.

Comment-4. In which condition the plants were grown? How many plants were used in this experiment?

Response: We add more detailed information to explain these in Materials and methods (2.1. Plant materials) in the revised manuscript.

Comment-5. <=     ??   did you mean  ≤; >=     ??   did you mean  ≥

Response: Thanks for pointing out this error. We have modified these (≤ and ≥) in Materials and methods (2.1. Plant materials) in the revised manuscript.

Comment-6. Discussion: the increases in CO2 fixation in the GE Arabidopsis and rice plants observed in this experiment is likely due to upregulation of the key genes in the carbon-fixation related pathways, because of overexpression of the EPSPS gene.        Is there any explanation why is this happening? What is the mechanism for upregulation of all those genes?

Response: Thanks for the comment. Regarding this comment, published results indicated that increased expression of the PPT gene can stimulate carbon fixation by increasing the content of aromatic amino acids, the number of fruits, and biomasses of plants. It is possible that more supply of PEP, which is from glycolysis starting from carbohydrate glucose, by PPT to EPSPS enzyme as substrate, will positively regulate the carbon fixation to produce more carbohydrates. The more fixed carbon was allocated into the shikimate pathway and consumed in EPSPS transgenic plants, the consumption of carbon fixation products possibly promoting the expression of key genes in the carbon-fixation related pathways. However, the relationship between PPT gene or PEP and carbon-fixation related genes is still unknown. Therefore, more studies are needed to further explore the true reasons.

Comment-7. The authors are overly optimistic and idealistic about the benefits of breeding EPSPS overexpressing crop and forest species, forgetting that GE plants are not allowed to be grown in much of the world, and the propagation of GE forest species would not be so simple.

Response: We agree with the reviewer’s comment that we are at the present overly optimistic about the benefits of breeding EPSPS overexpressing crop and forest species, due to the limitation of GE plants that are permitted to cultivated anywhere in the world. Therefore, we softened the statement of future applications of this technology by rewording some sentences in Discussion. However, given the fast development of the biotechnology and increasing growth of GE crops in the world, the potential of using GE plants to solve the world’s desperate problems of global warming may play its role in carbon neutrality.
